# Should LLMs Explain Every Time-Series Alert?
# A Reliability-Routed Audit for Structured Data Monitoring

**WANG Yibo** [1]   **Wang Shuai** [2]   **Zhang Ting** [3]   **Huang Runqing** [3]   **Tsin Hei Koo** [3]   **Cao Zhiqiang** [*][1]

## Abstract

Large language models can turn structured time-series alerts into fluent explanations, but fluency can amplify trust in weak or false alerts. We study a narrower question than anomaly detection: after a detector fires, should an LLM explanation be emitted at all? We audit three detector-side reliability signals: score extremeness, margin above threshold, and concentration of variable-level evidence. On SMD, conservative routing raises routed-alert precision from 0.125 to 0.512, but only at 0.011 coverage, showing that reliable explanation opportunities can be sparse under weak detector evidence. Machine-level validation/test splits expose high variance: across five splits, the hard router averages $0.169 \pm 0.126$ precision, with one split reaching 0.365 precision at 0.016 coverage. Across DeepSeek-v4-flash, DeepSeek-v4-pro, Qwen-plus, and Qwen-flash, routing reduces calls and improves an alert-validity proxy, whereas a 1000-alert neutral-prompt audit shows that LLM self-gating remains permissive across the same four models. This study contributes a reliability audit for LLM-assisted structured-data monitoring, focused on explanation eligibility after detector alerts.

## 1. Introduction and Positioning

Structured monitoring systems increasingly attach natural-language rationales to anomaly alerts. For multivariate time-series anomaly detection (TSAD), this is useful because operators want suspected root-cause variables, concise incident context, and scalar scores. The risk is that an LLM can produce a confident explanation for a weak or false alert. In alerting systems, such explanations can change which incidents are trusted, escalated, or ignored.

We study a narrow deployment question. The detector decides whether an alert exists; the explanation router decides whether an automated LLM rationale should be attached. When the router rejects an alert, the system preserves the alert record, withholds generated text, and logs insufficient evidence. The selective decision is applied to the explanation layer rather than to the alert itself.

The claim is an empirical audit claim. Numerical evidence already present in a structured-data monitoring pipeline can identify alerts whose detector-side support is too weak for automatic natural-language rationales. This audit exposes when a detector produces many alerts that require alert-level handling but lack enough support for generated incident text.

This paper contributes an empirical audit: a detector-side explanation router, precision–coverage and transfer analyses on SMD and TSB-UAD, and LLM direct/routed, self-gating, prompt-robustness, and cost checks. Together, these experiments frame explanation emission as a selective invocation problem for foundation-model-based monitoring.

**Positioning.** The analysis adopts precision–coverage language from selective prediction and reject-option classification (Chow, 1970; Geifman & El-Yaniv, 2017), but the decision object differs: the detector still emits an alert, while abstention applies only to the generated explanation attached to that alert. This places the evaluation on alert-pool accounting, abstention, and resource behavior for explanation emission.

TSAD benchmarks and detectors increasingly cover multivariate monitoring, calibrated scoring, and deep sequence models (Su et al., 2019; Xu et al., 2022; Wu et al., 2023; Alnegheimish et al., 2025; Liu et al., 2026a). Lightweight detectors provide a controlled way to ask whether cheap, broadly available detector-side reliability signals can support an explanation-emission audit. Stronger detectors may change the operating points, while retaining the monitoring question of whether an LLM rationale should be shown.

[1]Hangzhou International Innovation Institute, Beihang University, Hangzhou, China [2]North China University of Water Resources and Electric Power, Zhengzhou, China [3]Hong Kong University of Science and Technology, Hong Kong, China. Correspondence to: Cao Zhiqiang <zhiqiangcao@buaa.edu.cn>.

*Proceedings of the 2nd ICML Workshop on Foundation Models for Structured Data, Seoul, South Korea. 2026. Copyright 2026 by the author(s).*

Recent work also explores LLMs for time-series anomaly reasoning and diagnosis (Yang et al., 2025; Lan et al., 2025; Liu et al., 2026b; Devireddy & Huang, 2025). These studies motivate natural-language interfaces for structured monitoring. The focus here is the emission policy that governs when a monitoring system allows an LLM explanation. For foundation-model-based monitoring, explanation generation is an invocation decision: after a structured-data model or detector raises an alert, the LLM layer still needs a calibrated emission policy rather than a globally transferable default.

## 2. Reliability Routing

Let $x_t \in \mathbb{R}^d$ be a multivariate observation. A detector produces an anomaly score $s_t$, threshold $\tau$, and nonnegative variable evidence $e_{t,j}$. An alert is $a_t = \mathbf{1}[s_t \geq \tau]$. For each alert, the router computes

$$q_t = |S_{\text{train}}|^{-1} \sum_{s \in S_{\text{train}}} \mathbf{1}[s \leq s_t], \qquad (1)$$

$$m_t = \frac{s_t - \tau}{\max(|\tau|, \epsilon_s)}, \qquad (2)$$

$$c_t = \begin{cases} \dfrac{\max_j e_{t,j}}{\sum_j e_{t,j}}, & \sum_j e_{t,j} > 0, \\ 0, & \sum_j e_{t,j} = 0. \end{cases} \qquad (3)$$

Here $S_{\text{train}}$ is the training-score distribution for the same machine-detector pair, and $\epsilon_s$ is a small score-scale floor with the same units as $s_t$ and $\tau$. An explanation is emitted only if $a_t = 1$, $q_t \geq \alpha$, $m_t \geq \beta$, and $c_t \geq \gamma$.

The three signals have different failure modes. The quantile $q_t$ asks whether the score is extreme relative to the detector's training distribution. The margin $m_t$ asks whether the alert is safely above the operational threshold. The concentration $c_t$ asks whether variable-level evidence is localized enough to support a root-cause style explanation. A diffuse multivariate deviation may still deserve an alert, but it is a weak basis for a specific LLM rationale.

For robust_z, $e_{t,j}$ is robust per-variable deviation; for delta_z, first-difference deviation; for pca_residual, squared residual contribution. The hard router is the primary audit object because it is auditable and because learned gates can overfit validation machines; Table 3 evaluates them as baselines under the same inputs.

By default, an alert is a machine-detector-time tuple and all metrics are alert-level. Coverage is the fraction of detector alerts for which an explanation is emitted. For LLM text, an emitted explanation is counted when the response is parseable JSON with decision=ANOMALY and should_explain=true. Routed precision is the fraction of emitted explanations whose alert overlaps a ground-truth anomaly window, and false emit rate is one minus

routed precision. Because different LLMs abstain or fail parsing on different prompts, emitted-alert precision can vary across models on the same prompt set. This alert-validity proxy captures whether the underlying alert was real; semantic faithfulness, root-cause correctness, and operator usefulness require separate labels.

## 3. Experiments

**Data and detectors.** The experiments use SMD (Su et al., 2019) with 28 machines and three lightweight detectors: robust_z, delta_z, and pca_residual. The lightweight detectors keep the study focused on cheap, broadly available detector-side reliability signals for explanation routing. The full SMD detector-alert pool contains 219,017 alerts with raw alert precision about 0.1005, which makes it a stress test for explanation emission. A routing-only sweep on six labeled TSB-UAD univariate files provides a second public-dataset sanity check. Because this TSB-UAD subset is univariate, concentration serves as a limited localization signal.

**Router sweep and ablation.** The router sweep varies score quantile, margin, and concentration thresholds. On SMD, a permissive point (0.995, 0.05, 0.35) has routed precision 0.125 at 0.503 coverage. A conservative point (0.9995, 10.0, 0.70) has precision 0.512 at 0.011 coverage. Signal ablations disable subsets of the three conditions and select the highest-precision point with at least 1% coverage. This tests whether the router is only a disguised score threshold or whether concentration and margin add information.

**Validation/test split.** To test post-hoc threshold selection, SMD is split by machine using stratification over detector-alert prevalence. The main validation split contains 109,799 alerts and the held-out split contains 109,218 alerts; both have raw precision approximately 0.1005. Thresholds and learned gates are selected only on validation machines and reported on held-out machines. Learned gates use the same three signals: logistic regression over $(q, m, c)$, a depth-2 decision tree, and score-quantile ranking. An auxiliary variance check repeats this machine-level split over five seeds.

**LLM audit.** We evaluate DeepSeek-v4-flash, DeepSeek-v4-pro, Qwen-plus, and Qwen-flash at temperature 0 with JSON output. The explanation subset has 213 SMD detector-alert prompts: 80 router-allowed true alerts, 53 router-allowed false alerts, and 80 router-rejected alerts. This subset supports controlled model comparison; Table 1 reports full-pool routing coverage. Direct sends every detector-alert prompt under the same JSON contract; routed LLM sends only router-allowed prompts.

The audit logs request counts, token usage, wall-clock latency, and also constructs an independent 1000-alert SMD sample for a neutral-prompt LLM self-gating audit. The sample excludes the enriched 213-prompt audit subset and earlier development samples.

# 4. Results

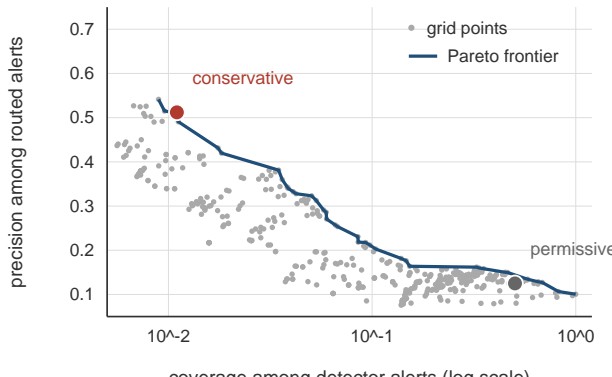

*Figure 1.* SMD router sweep. Conservative routing improves routed precision from 0.125 to 0.512 but reduces coverage from 0.503 to 0.011.

Figure 1 and Table 1 show the central tradeoff. On SMD, a conservative router moves routed precision from near raw detector precision to 0.512, but explains only 1.1% of detector alerts and still admits many false alerts. On TSB-UAD, a high-precision point is possible, but the SMD conservative threshold transfers poorly. The TSB-UAD SMD-transfer row is a negative transfer check: thresholds selected on SMD select almost no valid TSB-UAD alerts, motivating dataset-specific or online calibration rather than static cross-dataset threshold transfer. Routing therefore serves as an audit mechanism for exposing precision-coverage and transfer behavior. Appendix B reports a compact USAD-style reconstruction-detector sanity check on SMD; under this reconstruction detector, validation-selected routing improves held-out routed precision to 0.600 at 0.0286 coverage.

*Table 1.* Selected router operating points.

| Data | Point | Prec. | Cov. | False emit |
|------|-------|-------|------|-----------|
| SMD | permissive | 0.125 | 0.503 | 0.875 |
| SMD | conservative | 0.512 | 0.011 | 0.488 |
| TSB-UAD | permissive | 0.716 | 0.608 | 0.284 |
| TSB-UAD | high-precision | 0.992 | 0.199 | 0.008 |
| TSB-UAD | SMD-transfer | 0.000 | 0.001 | 1.000 |

**Signal ablation.** Table 2 shows that margin and concentration add information beyond score quantile alone. Score-

*Table 2.* SMD signal ablation, selecting the best precision point with coverage $\geq 1\%$.

| Router | Prec. | Cov. | False emit |
|--------|-------|------|-----------|
| Score only | 0.158 | 0.353 | 0.842 |
| Margin only | 0.160 | 0.334 | 0.840 |
| Score+margin | 0.162 | 0.326 | 0.838 |
| Score+concentration | 0.277 | 0.019 | 0.723 |
| Margin+concentration | 0.421 | 0.013 | 0.579 |
| Full router | 0.512 | 0.011 | 0.488 |

*Table 3.* SMD learned-gate and hard-router baselines selected on validation machines and evaluated on held-out machines. Main split shown; full five-seed statistics appear in Appendix A.

| Policy | Val. P | Val. C | Test P | Test C |
|--------|--------|--------|--------|--------|
| Hard router grid | 0.470 | 0.014 | 0.365 | 0.016 |
| Logistic(q,m,c) | 0.975 | 0.010 | 0.144 | 0.020 |
| Depth-2 tree(q,m,c) | 0.381 | 0.010 | 0.074 | 0.048 |
| Score-quantile rank | 0.239 | 0.010 | 0.112 | 0.274 |

only and margin-only routers stay near 0.16 precision even while covering one third of alerts. Concentration appears useful mainly when paired with margin or score, and the full three-signal router reaches the highest precision under the same minimum-coverage constraint. The cost is visible: useful concentration thresholds also remove most alerts.

Table 3 shows the main validation/test split. Validation and test machines are balanced for raw alert precision, yet selected policies degrade on held-out machines. The logistic learned gate reaches 0.975 validation precision at 1% validation coverage, but only 0.144 test precision. A five-seed check confirms high variance: test precision mean $\pm$ std is $0.169 \pm 0.126$ for the hard router, $0.443 \pm 0.425$ for logistic, $0.178 \pm 0.184$ for the tree, and $0.137 \pm 0.042$ for score-rank; test coverage mean $\pm$ std is $0.161 \pm 0.186$, $0.013 \pm 0.010$, $0.100 \pm 0.125$, and $0.226 \pm 0.093$. The hard router's high mean test coverage reflects threshold transfer drift: only validation coverage is constrained. Score-rank's high held-out coverage reflects over-transfer of a one-signal score-quantile cutoff: it covers many more held-out alerts while sacrificing precision. Thus Table 3 provides evidence of unstable policy selection across held-out machines.

**LLM audit and self-gating.** Table 4 summarizes the enriched 213-prompt LLM audit. Routing improves the emitted-alert validity proxy for all four models. Read the 62.4% call rate as sampled-audit behavior; Table 1 gives the corresponding full-pool SMD coverage of 0.011. In the enriched 213-prompt subset, the smallest precision gain, for DeepSeek-v4-flash, has a bootstrap interval crossing zero, so we treat this subset audit as directional evidence.

*Table 4.* LLM explanation audit on an enriched 213-prompt SMD subset. "Emit" counts model-emitted explanations; "Prec." is emitted-alert validity. Routed requests are sampled-subset call rates; Table 1 reports full-pool coverage. DS-v4-flash and DS-v4-pro abbreviate DeepSeek-v4-flash and DeepSeek-v4-pro.

| Model | Direct emit | Direct P | Routed sent | Routed emit/sent | Routed P |
|---|---|---|---|---|---|
| DS-v4-flash | 188/213 | 0.559 | 133/213 | 125/133 | 0.600 |
| DS-v4-pro | 173/213 | 0.566 | 133/213 | 119/133 | 0.613 |
| Qwen-plus | 207/213 | 0.517 | 133/213 | 133/133 | 0.602 |
| Qwen-flash | 206/213 | 0.515 | 133/213 | 133/133 | 0.602 |

Appendix E adds a population-weighted 800-prompt robustness audit; after weighting by full SMD alert-pool strata, routed precision remains 0.512–0.539 versus 0.101–0.122 for direct prompting, with gain intervals excluding zero.

Table 5 shows the complementary result: model-side self-gating remains permissive, allowing 55.5–69.5% of alerts and reaching only 0.124–0.128 precision. This is only a modest gain over raw alert-pool precision, so self-gating is a weak substitute for detector-side numerical routing. Appendix F varies the self-gating prompt and shows that model-side gates are prompt-sensitive: stricter prompts can reduce allow rate, while minimal or calibration-aware prompts can remain highly permissive.

*Table 5.* Neutral-prompt LLM self-gating on an independent 1000-alert SMD sample. Raw full-pool alert precision is 0.1005. DS-v4-flash and DS-v4-pro abbreviate DeepSeek-v4-flash and DeepSeek-v4-pro.

| Model | Allowed | Allow rate | Prec. | False allowed |
|---|---|---|---|---|
| DS-v4-flash | 569/1000 | 0.569 | 0.128 | 496 |
| DS-v4-pro | 555/1000 | 0.555 | 0.128 | 484 |
| Qwen-plus | 695/1000 | 0.695 | 0.124 | 609 |
| Qwen-flash | 672/1000 | 0.672 | 0.125 | 588 |

The cost/latency result is a deployment accounting check: fewer routed requests reduce token use and mean per-alert latency. Mean per-alert latency drops from 13.42s to 9.98s for DeepSeek-v4-flash, from 76.36s to 40.71s for DeepSeek-v4-pro, from 7.14s to 4.39s for Qwen-plus, and from 4.54s to 2.61s for Qwen-flash; DeepSeek-v4-flash uses a 100-alert timing subset under the same routing policy.

*Table 6.* Cost and latency checks for direct versus routed prompting. Mean latency is averaged over detector-alert prompts; routed-abstained prompts incur only local routing cost. DeepSeek-v4-flash latency uses $n = 100$; all other rows use the 213-prompt audit subset. DS-v4-flash and DS-v4-pro abbreviate DeepSeek-v4-flash and DeepSeek-v4-pro.

| Model | Requests | Tokens | Mean latency |
|---|---|---|---|
| DS-v4-flash | 100→62 | 341,689→212,198 | 13.42s→9.98s |
| DS-v4-pro | 213→133 | 873,135→544,676 | 76.36s→40.71s |
| Qwen-plus | 213→133 | 761,252→474,770 | 7.14s→4.39s |
| Qwen-flash | 213→133 | 759,836→473,494 | 4.54s→2.61s |

## 5. Discussion and Limitations

**Practical implications.** Evaluation should report alert counts, raw precision, routed coverage, emitted-explanation definitions, validation/test protocols, direct and self-gated baselines, and cost/latency accounting.

**Transfer and calibration.** Detector-side numerical evidence reveals sparse explanation opportunities under weak alert evidence. Thresholds transfer brittlely across machines and datasets, and learned gates can overfit even when validation/test splits are balanced by raw alert precision. Appendix C further shows that raising validation coverage targets to 5–10% drives held-out precision back toward the raw alert-pool baseline, exposing the precision cost of broader explanation coverage. Practical systems therefore need calibration by device family, detector family, and operational coverage target, rather than a single global explanation threshold.

**Limitations.** The main evaluation uses lightweight detectors; Appendix B provides a compact USAD-style reconstruction sanity check that exhibits the same audit pattern. Full-scale evaluations with stronger TSAD architectures such as Anomaly Transformer, TimesNet, TranAD, or production-grade USAD variants remain natural follow-ups. The explanation metric remains an alert-validity proxy; Appendix D adds only a variable-overlap diagnostic, showing that named variables largely track detector top-features rather than independent semantic faithfulness. The TSB-UAD check is univariate, while multivariate root-cause evaluation requires multivariate labels. The LLM experiments use deterministic decoding; Appendix F shows that self-gating behavior is prompt-sensitive, so prompt policy remains part of the calibration problem.

**Conclusion.** LLM explanation emission for time-series alerts should be selective by default. For foundation-model-based monitoring systems, detector-side reliability signals provide a lightweight, auditable router before costly LLM explanation calls; explanation eligibility should be calibrated, measured, and reported rather than treated as a default post-processing step.

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

## A. Five-Seed Machine Split Details

Table 7 reports the full auxiliary split check used to summarize validation/test instability in the Results section. Learned gates and score ranking choose cutoffs near a 1% validation-coverage target. For the hard-router grid, the selected point is the highest-precision validation point satisfying the minimum coverage constraint, which can yield higher validation coverage when the grid is coarse. All rows then report held-out machines with thresholds fixed after validation.

*Table 7.* Five machine-level split seeds for the SMD routing baselines.

| Seed | Policy | Val. P | Val. C | Test P | Test C |
|---|---|---|---|---|---|
| 7 | Hard router | 0.470 | 0.014 | 0.365 | 0.016 |
| 7 | Logistic | 0.975 | 0.010 | 0.144 | 0.020 |
| 7 | Tree | 0.381 | 0.010 | 0.074 | 0.048 |
| 7 | Score rank | 0.239 | 0.010 | 0.112 | 0.274 |
| 17 | Hard router | 0.292 | 0.140 | 0.095 | 0.438 |
| 17 | Logistic | 0.974 | 0.010 | 0.116 | 0.021 |
| 17 | Tree | 0.848 | 0.010 | 0.068 | 0.319 |
| 17 | Score rank | 0.380 | 0.010 | 0.101 | 0.315 |
| 27 | Hard router | 0.585 | 0.011 | 0.099 | 0.073 |
| 27 | Logistic | 0.949 | 0.010 | 0.141 | 0.018 |
| 27 | Tree | 0.179 | 0.010 | 0.182 | 0.061 |
| 27 | Score rank | 0.142 | 0.010 | 0.159 | 0.151 |
| 37 | Hard router | 0.511 | 0.015 | 0.062 | 0.266 |
| 37 | Logistic | 0.453 | 0.010 | 0.948 | 0.002 |
| 37 | Tree | 0.349 | 0.010 | 0.495 | 0.003 |
| 37 | Score rank | 0.170 | 0.010 | 0.114 | 0.285 |
| 47 | Hard router | 0.536 | 0.017 | 0.227 | 0.012 |
| 47 | Logistic | 0.454 | 0.010 | 0.866 | 0.001 |
| 47 | Tree | 0.698 | 0.010 | 0.069 | 0.072 |
| 47 | Score rank | 0.342 | 0.010 | 0.200 | 0.103 |

## B. Reconstruction-Style Deep Detector Sanity Check

Appendix B reports a routing-only sanity check with a compact USAD-style reconstruction detector. The detector score is the summed squared reconstruction error, and variable evidence is the per-variable squared reconstruction residual. The detector threshold is selected from train scores only. This check tests whether the router still exposes a precision–coverage tradeoff outside the lightweight detector pool; it is not a SOTA TSAD comparison.

*Table 8.* USAD-style reconstruction-detector routing sanity check on SMD. The detector threshold uses the 0.995 train-score quantile; router operating points are selected on validation machines and reported on held-out machines.

| Point | Raw P | Alerts | Routed P | Coverage | False emit |
|---|---|---|---|---|---|
| All-pool raw | 0.1523 | 82,678 | 0.1523 | 1.0000 | 0.8477 |
| Held-out raw | 0.1059 | 54,839 | 0.1059 | 1.0000 | 0.8941 |
| Best val hard / 1–5% | 0.1059 | 54,839 | 0.6000 | 0.0286 | 0.4000 |
| Val target 10% | 0.1059 | 54,839 | 0.5875 | 0.0369 | 0.4125 |

The 1% and 5% validation-coverage targets select the same hard-router point because the threshold grid is coarse and the rule maximizes validation precision subject to a minimum coverage constraint.

## C. Coverage-Targeted SMD Operating Points

Appendix C reports validation-selected SMD operating points at fixed minimum coverage targets. For each target, the analysis uses the same SMD alert features and machine-level validation/test split as the main analysis. The hard-router threshold triple is selected on validation machines by maximizing validation precision subject to minimum validation coverage, then evaluated on held-out machines. The validation and held-out splits contain 109,799 and 109,218 detector alerts, respectively, with raw held-out precision 0.1005 and finite routing signals. These rows expose the precision cost of higher explanation coverage; they do not change the main operating point.

*Table 9.* Coverage-targeted SMD lightweight-router operating points. Operating points are selected on validation machines by maximizing precision subject to minimum validation coverage, then evaluated on held-out machines.

| Target | $q$ | $m$ | $c$ | Val P | Val C | Test P | Test C | False emit |
|---|---|---|---|---|---|---|---|---|
| 1% | 0.9999 | 5.00 | 0.70 | 0.4703 | 0.0137 | 0.3645 | 0.0160 | 0.6355 |
| 5% | 0.9995 | 0.05 | 0.00 | 0.2480 | 0.2359 | 0.1074 | 0.4445 | 0.8926 |
| 10% | 0.9995 | 0.05 | 0.00 | 0.2480 | 0.2359 | 0.1074 | 0.4445 | 0.8926 |

The 5% and 10% targets select the same operating point under the existing grid; at these higher targets, held-out precision falls close to the raw held-out precision of 0.1005.

## D. Automatic Variable-Label Overlap Audit

Appendix D adds an automatic variable-label overlap audit on emitted true-alert explanations using SMD interpretation labels. The audit compares model-named root-cause variables with SMD anomalous-variable labels and reports Hit@1, Hit@3, and overlap F1. This is a variable-overlap diagnostic rather than a semantic faithfulness, causal correctness, or operator-usefulness evaluation. Because prompts expose detector top-features, the audit also compares each LLM output with a top-feature baseline.

*Table 10.* Automatic variable-label overlap audit on emitted true-alert explanations. TopFeat uses detector top-features exposed in the prompt.

| Model | Strategy | Usable | Hit@3 | LLM F1 | TopFeat F1 |
|---|---|---|---|---|---|
| DS-v4-flash | direct | 105 | 0.981 | 0.402 | 0.414 |
| DS-v4-flash | routed | 75 | 0.960 | 0.373 | 0.396 |
| DS-v4-pro | direct | 98 | 0.959 | 0.408 | 0.412 |
| DS-v4-pro | routed | 73 | 0.973 | 0.358 | 0.394 |
| Qwen-plus | direct | 107 | 0.972 | 0.375 | 0.414 |
| Qwen-plus | routed | 80 | 0.975 | 0.358 | 0.395 |
| Qwen-flash | direct | 106 | 0.962 | 0.387 | 0.413 |
| Qwen-flash | routed | 80 | 0.963 | 0.366 | 0.395 |

Across all model/strategy pairs, copying diagnostics show that predicted variables are subsets of prompt-exposed top-features. This supports a limited conclusion: generated root-cause fields preserve detector evidence, but this audit does not demonstrate independent causal reasoning or semantic faithfulness.

## E. Population-Weighted Expanded LLM Audit

Appendix E reports an expanded 800-prompt direct/routed audit. The sample is intentionally balanced by router outcome and alert validity, so unweighted direct/routed precision is a balanced-sample diagnostic rather than a full-pool estimate. Each stratum is therefore weighted by its frequency in the full SMD detector-alert pool.

*Table 11.* Full-pool stratum weights for the expanded 800-prompt audit.

| Stratum | Full count | Sample count | Weight |
|---|---|---|---|
| Allowed true | 1,183 | 250 | 4.732 |
| Allowed false | 1,129 | 250 | 4.516 |
| Rejected true | 20,829 | 150 | 138.860 |
| Rejected false | 195,876 | 150 | 1305.840 |

*Table 12.* Population-weighted direct/routed audit on the expanded 800-prompt sample. Unweighted metrics are not full-pool estimates because the sample is balanced by route outcome and alert validity.

| Model | Direct P | Routed P | Gain | 95% CI | Parse fail |
|---|---|---|---|---|---|
| DS-v4-flash | 0.115 | 0.528 | 0.413 | [0.396, 0.428] | 0 |
| DS-v4-pro | 0.122 | 0.539 | 0.417 | [0.397, 0.435] | 8 |
| Qwen-plus | 0.110 | 0.512 | 0.401 | [0.394, 0.408] | 0 |
| Qwen-flash | 0.101 | 0.512 | 0.411 | 0.411 (deg.) | 0 |

DeepSeek-v4-pro uses a JSON-enforced rerun in Table 12; its strict parse/API failures fall to 8 rows. Qwen-flash has a degenerate stratified-bootstrap interval because it emits every direct and routed row under fixed-stratum resampling. This table is a robustness diagnostic rather than a replacement for the cleaner 213-prompt main audit.

## F. Prompt Robustness for LLM Self-Gating

Appendix F varies the self-gating prompt on the same independent 1000-alert SMD sample. The strict prompt asks the model to allow only jointly strong reliability evidence; the calibration-aware prompt reminds the model that raw alert-pool precision is about 0.1005; the minimal prompt gives the same signals with minimal policy wording. This experiment tests prompt sensitivity of model-side self-gating, not explanation quality.

*Table 13.* Prompt robustness for the 1000-alert LLM self-gating audit. DS-v4-pro uses the JSON-enforced rerun; 'cal.-aware' abbreviates calibration-aware.

| Model | Prompt | Allow | Prec. | False allowed | Parse fail |
|---|---|---|---|---|---|
| DS-v4-flash | strict | 0.183 | 0.169 | 152 | 8 |
| DS-v4-pro | strict | 0.180 | 0.156 | 152 | 0 |
| Qwen-plus | strict | 0.303 | 0.129 | 264 | 0 |
| Qwen-flash | strict | 0.248 | 0.129 | 216 | 0 |
| DS-v4-flash | cal.-aware | 0.885 | 0.103 | 794 | 3 |
| DS-v4-pro | cal.-aware | 0.944 | 0.100 | 850 | 0 |
| Qwen-plus | cal.-aware | 0.887 | 0.103 | 796 | 0 |
| Qwen-flash | cal.-aware | 1.000 | 0.100 | 900 | 0 |
| DS-v4-flash | minimal | 0.980 | 0.100 | 882 | 0 |
| DS-v4-pro | minimal | 0.960 | 0.103 | 861 | 0 |
| Qwen-plus | minimal | 1.000 | 0.100 | 900 | 0 |
| Qwen-flash | minimal | 1.000 | 0.100 | 900 | 0 |

The prompt variants show that self-gating is policy-sensitive. Strict prompts can reduce allow rate, but minimal or calibration-aware prompts can remain highly permissive and near raw alert-pool precision. This reinforces that explanation emission needs explicit calibration rather than model-side self-regulation.

