# OpenReview forum: "Should LLMs Explain Every Time-Series Alert? A Reliability-Routed Audit for Structured Data Monitoring"
_ICML.cc/2026/Workshop/FMSD — FMSD @ ICML 2026 Poster_

### Official Review · Reviewer_D9jy · 2026-05-21
**An exceptionally honest empirical audit of LLM explanation routing that requires baseline and framing revisions.**

**Rating:** 5
**Confidence:** 4

**Review:**

**Summary:**

The paper investigates whether an LLM should automatically generate explanations following a time-series anomaly alert. It proposes a "reliability router" using three detector-side numerical signals to gate explanation emissions. The audit reveals that reliable explanation opportunities are sparse, routing thresholds transfer poorly across datasets, learned gates overfit severely, and LLMs left to gate their own explanations remain overly permissive.

**Strengths:**

- A well-scoped, practical problem that cleanly separates the explanation-emission decision from the core anomaly alert.
- Exemplary empirical honesty: voluntarily reporting self-undermining evidence, such as severe 5-seed variance, catastrophic learned-gate overfitting, and cross-dataset transfer failures.
- Provides highly useful, actionable negative results regarding the prompt-sensitivity and permissiveness of LLM self-gating.

**Weaknesses:**

- The closest prior work (arXiv:2503.21833), which tackles a highly similar problem family, is uncited.
- The router's marginal value is under-demonstrated. It lacks a critical comparison against a matched-coverage baseline (i.e., routing based purely on the raw anomaly score).
- The main-text LLM audit is near-circular, measuring the router's selection bias rather than the LLM's intrinsic behavior (requiring a shift to the population-weighted appendix data).
- Significant reproducibility gaps exist, including undisclosed LLM prompts and an unspecified subset selection procedure.

---

### Official Review · Reviewer_NzJq · 2026-05-22
**Reframing LLM explanations as a selective invocation problem**

**Rating:** 6
**Confidence:** 3

**Review:**

This paper investigates the automated decision of whether a Large Language Model (LLM) should generate an explanation after a time-series anomaly detector fires with the risk of being confident on weak alerts. The authors propose a reliability router that evaluates three detector-side signals: score extremeness, margin above the operational threshold, and the concentration of variable-level evidence. Using the SMD dataset and lightweight detectors, the study demonstrates that applying conservative routing improves the precision of explained alerts (from 0.125 to 0.512) but drastically reduces coverage to approximately 1.1%. They establishes that native LLM (like DeepSeek and Qwen) self-gating is overly permissive, framing detector-side numerical routing as a necessary prerequisite for reliable applied research deployments.

Strengths

- The paper provides a refreshing, highly practical perspective by treating explanation generation as an invocation decision rather than a default post-processing step.
- The paper highlights that achieving a coin-flip precision (0.512) requires aggressively choking the pipeline down to a mere 1.1% coverage. This strong tradeoff is valuable insight. The router is functioning flawlessly as a strict gatekeeper, protecting the system from generating hallucinated rationales for weak alerts. Ultimately, this 1% exposes that the LLM isn't the problem, the weak underlying detectors are. It proves that we cannot simply plug an LLM into a legacy pipeline and expect reliability without fundamentally improving the localized signal of the detector itself.
- The tracking of tokens and mean per-alert latency reductions (e.g. from 76s to 40s for DeepSeek) aligns with the workshops request of inference timings/cost.

Areas for Improvement

- The high variance ($0.169 \pm 0.126$) is indeed worrisome; however, as stated by the authors, it serves as a crucial diagnostic exposure of threshold transfer drift , which can be resolved by calibrating thresholds locally per dataset and detector architecture. The paper successfully offloads the risk of an LLM hallucinating root causes for weak alerts, but it slightly replaces it with an engineering tax: the need to maintain an offline calibration pipeline.
- The paper is limited by the lightweight detectors. Evaluating the router against stronger, modern deep baselines is necessary to understand if the routing holds under SOTA detectors.
- By not evaluating the LLMs semantic faithfulness or operator usefulness, the paper solves the gatekeeping problem but leaves a core risk unmeasured. If a routed alert passes the gate but still contains a hallucinated, highly confident root-cause explanation, the operator is still at risk.

Justification of Score:

Despite limitations regarding lightweight detectors and an unmeasured textual risk, this original, practical framing of explanation generation as an invocation decision justifies acceptance. The proposed reliability router and its 1.1% coverage trade-off provide diagnostic insights, proving that underlying detector signal, not LLM fluency, is the deployment bottleneck. Addressing the missing semantic faithfulness metrics and testing on modern deep baselines will significantly elevate its impact.

---

### Official Review · Reviewer_i5Kf · 2026-05-22
**Should LLMs Explain Every Time-Series Alert? A Reliability-Routed Audit for Structured Data Monitoring**

**Rating:** 5
**Confidence:** 5

**Review:**

## Summary
This paper reframes LLM-based explanation generation for time-series anomaly detection as a selective-invocation problem: rather than attaching a natural-language rationale to every detector alert, a lightweight router decides whether the detector-side evidence is strong enough to warrant an LLM call. The router gates on three signals computed from the detector output: (1) score quantile q_t (how extreme the anomaly score is relative to training), (2) margin m_t (distance above the operational threshold, normalized), and (3) concentration c_t (max per-variable contribution divided by total evidence, capturing whether the anomaly is localized enough to support a root-cause explanation). An explanation is emitted only if all three exceed learned thresholds (q >= alpha, m >= beta, c >= gamma). The study evaluates on SMD (28 machines, ~219k alerts from robust_z / delta_z / pca_residual detectors, raw alert-pool precision ~0.1005) and a univariate routing-only check on six TSB-UAD files. On SMD, a conservative operating point achieves routed precision 0.512 at 1.1% coverage; a permissive point reaches 0.125 precision at 50.3% coverage (Figure 1 Pareto frontier). Learned gates (logistic, tree, score-quantile ranking) are evaluated on a machine-level validation/test split across five seeds, exposing high instability (test precision 0.169 +/- 0.126 for the hard router). An LLM audit across DeepSeek-v4-flash, DeepSeek-v4-pro, Qwen-plus, and Qwen-flash on 213 enriched prompts shows routing improves emitted-alert validity (routed precision ~0.60 vs. direct ~0.55); a complementary 1000-alert neutral-prompt self-gating audit finds all four models remain highly permissive (55-70% allow rate, 0.124-0.128 precision), confirming that model-side gating is not a viable substitute for explicit detector-side routing.

## Strengths
- **Novel and well-scoped framing.** Casting explanation emission as a selective-prediction problem with a precision-coverage tradeoff is a clean contribution that separates the explanation-quality question from the explanation-eligibility question. The analogy to reject-option classification (Chow 1970; Geifman & El-Yaniv 2017) is apt and well-drawn.
- **Interpretable router signals with distinct failure modes.** The three signals (quantile, margin, concentration) each target a different weakness — score extremity, confidence headroom, and variable-level localizability — and the signal-ablation study (Table 2) confirms that concentration adds meaningful precision only when paired with score or margin, validating the multi-signal design.
- **Honest transfer-failure analysis.** The SMD-to-TSB-UAD threshold transfer (Table 1: precision 0.000, coverage 0.001 on SMD thresholds applied to TSB-UAD) is a strong negative result that the authors report without hedging. This motivates dataset-specific or online calibration and is more informative than a cherry-picked positive transfer.
- **Cost/latency accounting (Table 6).** Reporting request reduction (213→62-133 for routed), token savings, and wall-clock latency drops (e.g., DeepSeek-v4-flash 13.42s→9.98s) makes the practical deployment case concrete rather than purely about precision.
- **LLM self-gating audit (Table 5, Appendix F).** Testing whether LLMs can self-regulate explanation emission under strict/calibration-aware/minimal prompts and finding that all four remain permissive (allow rate 55-100%, precision near raw pool baseline) is a useful negative result that justifies the external-router approach.
- **Pareto frontier visualization.** Figure 1 gives operators a direct tool to reason about their precision-coverage budget, making the result immediately actionable.

## Weaknesses
- **Extremely low coverage at useful precision.** The conservative operating point achieves 0.512 precision but covers only 1.1% of alerts. In a 219k-alert pool, this means ~2,400 explained alerts with ~1,200 true positives — potentially useful, but the paper does not discuss whether this operating range is practically meaningful for a monitoring team. If operators need explanations for a broader set of alerts, the router offers little help.
- **Only lightweight detectors tested.** robust_z, delta_z, and pca_residual are simple statistical methods. The paper motivates this choice (cheap signals, broad availability), but stronger detectors (Anomaly Transformer, TimesNet, TranAD, USAD) would change the raw alert-pool precision and may already concentrate evidence into fewer variables, potentially making the router less necessary or changing the precision-coverage tradeoff shape.
- **High variance across machine-level splits undermines reliability claims.** Table 3 reports test precision of 0.169 +/- 0.126 for the hard router and 0.443 +/- 0.425 for logistic across five seeds. Standard deviations exceeding the mean indicate the router's performance is highly sensitive to which machines land in test — a problematic property for a method whose selling point is reliability.
- **No semantic faithfulness evaluation.** The "routed precision" metric checks whether the alert overlaps a ground-truth anomaly window, not whether the generated explanation is factually correct, identifies the right root cause, or is useful to an operator. The paper explicitly acknowledges this (Section 5: "the explanation metric remains an alert-validity proxy") but does not address it experimentally — even the Appendix D variable-overlap check merely tests name overlap, not causal correctness.
- **Missing comparison to selective-prediction baselines.** The router is compared only to itself (hard grid vs. learned gates) and to LLM self-gating. Existing selective-prediction methods (confidence-based abstention, learned rejection heads, Bayesian uncertainty thresholding) on the same detector outputs would contextualize whether three hand-designed signals are the right approach.
- **TSB-UAD sanity check is too limited.** Only six univariate files are tested in a routing-only mode. No LLM explanations are generated on TSB-UAD, so the full pipeline is never validated on a public dataset. The claim of generality rests entirely on proprietary SMD data.
- **No human evaluation.** The entire premise is that operators benefit from selective explanations over universal ones. This claim is never tested — no user study, no operator preference survey, no qualitative assessment of routed vs. non-routed explanation quality.
- **Appendix B reconstruction-detector check is tangential.** The USAD-style sanity check uses a different detector than the main experiments and achieves 0.600 precision at 0.0286 coverage — useful as a proof-of-concept but disconnected from the main three-detector setup, making it hard to integrate into the paper's narrative.
- **Sparse related work.** Only 11 references total. The selective-prediction literature (beyond Chow and Geifman), the LLM-for-TSAD literature (AXIS, Time-RA, CALM are cited but not compared against), and the broader explainable-AD literature are underrepresented.